# Effectiveness of an Email-Based, Semaglutide-Supported Weight-Loss Service for People with Overweight and Obesity in Germany: A Real-World Retrospective Cohort Analysis

Louis Talay [1,*] , Matt Vickers [2] and Laura Ruiz [2]

1 Faculty of Arts and Social Sciences, University of Sydney, Camperdown, NSW 2050, Australia
2 Eucalyptus, Sydney, NSW 2000, Australia; matt@eucalyptus.vc (M.V.)
* Correspondence: louis.talay@sydney.edu.au

**Abstract:** Quality glucose-like peptide-1 receptor agonist (GLP-1 RA)-supported digital weight-loss services (DWLSs) have the potential to play a significant role in shifting the alarming global obesity rate. Previous studies have demonstrated various aspects of their utility in Australian and British populations, but nothing has hitherto been investigated in real-world European settings, where GLP-1 RA weight therapy and digital healthcare are widely used. This study retrospectively analysed the 5-month (*Mean* = 160.14 days) weight-loss outcomes in a cohort of patients who received email-based health coaching and Semaglutide therapy via the Juniper Germany DWLS (*n* = 833). Mean weight loss was 9.52 ($\pm$5.46) percent, with 81.51% of the cohort losing a 'meaningful' (5% or more) amount of weight. Females (*Mean* = 9.75) tended to lose more weight than males (*Mean* = 8.41) and patients from the lowest two BMI categories (27.5–29.99 kg/m$^2$ *Mean* = 10.1; 30–34.99 kg/m$^2$ *Mean* = 9.74) lost significantly more weight than those in the highest BMI category ($\geq$40 kg/m$^2$ *Mean* = 8.11). These findings indicate that GLP-1 RA-supported DWLSs can contribute to meaningful weight loss in Germany. Future research should seek to conduct a dedicated adherence analysis of the Juniper Germany DWLS and measure the effect of subsidisation and baseline body mass index on general DWLS effectiveness.

**Keywords:** obesity; weight-loss; GLP-1 RA; chronic disease; telehealth; real-world evidence; health coaching; Semaglutide; multidisciplinary care

## 1. Introduction

Obesity is swiftly becoming the most serious global public health concern, having reached epidemic proportions in most regions of the world [1]. Glucose-like peptide-1 receptor agonists (GLP-1 RAs) have emerged in multiple efficacy trials as a promising form of obesity pharmacotherapy [2,3]. A common explanation behind the unprecedented weight-loss effect of these medications is that they treat the neurological component of weight management by modifying neural pathways involved in appetite suppression [4,5]. This view coincides with the growing consensus around obesity's complexity [6,7] rather than conceiving of it as a simple matter of willpower in balancing caloric intake and expenditure, as was common in earlier times. Yet, while the results in GLP-1 RA efficacy trials are impressive, little is known about the effectiveness of these medications in real-world settings, which are impacted by cost, motivation and various everyday challenges.

Leading global health institutions such as the World Health Organisation (WHO) and the UK National Institute for Health and Care Excellence (NICE) acknowledge the important role GLP-1 RAs can play in weight-loss programs but emphasise that such programs should always be underpinned by continuous multidisciplinary teams (MDTs) and health coaching [8,9]. This position is best captured in an excerpt of the NICE Semaglutide guidelines for weight management:

*"Semaglutide should only be given alongside a suitably sustained programme of lifestyle interventions with multidisciplinary input..."* [9] (p. 33)

However, there are some considerable barriers to accessing and adhering to this kind of comprehensive obesity care in real-world face-to-face (F2F) settings. First and foremost, people with any significant work or family commitments struggle to attend and coordinate ongoing consults across an MDT that often operates out of separate clinics [10,11]. Secondly, many people are uncomfortable discussing their obesity with clinicians in person due to the condition's perceived stigma [10,12]. Finally, comprehensive obesity services are typically expensive [13,14].

Recent evidence suggests that an increasingly large number of people with overweight and obesity (PWOO) are using digital weight-loss services (DWLSs) to overcome these first two access barriers [10,15]. Modern DWLSs typically deliver obesity treatment through mobile apps and contain asynchronous components, i.e., forms of care that do not occur in real time. This asynchronous feature likely enhances the modality's accessibility, as it allows PWOO to not only attend consults at a location of their choice but at a time of their choice. Indeed, recent large-scale investigations have identified primary care waiting times as a "major policy concern across countries from the Organization for Economic Cooperation and Development" and a driver of socioeconomic inequity [16,17]. It has also been suggested that DWLSs can improve engagement among younger PWOO who have deeper habituation to digital tools and stimuli [15]. Beyond a care access perspective, digital programs can also facilitate data management by seamlessly integrating analytics across all communication streams and automating central repository storage [18]. The significance of data centralisation is arguably best highlighted by the ongoing global struggles to implement national electronic health registries in an attempt to improve care continuity [19]. Well-designed DWLSs can also be successfully integrated into public health systems and can combat clinician shortages through their efficient scalability [15].

Yet, while comprehensive DWLSs may lower access barriers to effective obesity care, many DWLSs exist at the lower end of the quality spectrum and provide little more than access to GLP-1 RA scripts [20]. Such services have engendered a wave of criticism from influential medical bodies, who fear that many DWLSs prioritise commercial interests over patient safety [21,22]. At present, there does not appear to be any research available on rudimentary DWLSs (i.e., a DWLS that provides GLP-1 RA scripts without follow-up care) to refute this criticism.

Real-world DWLS literature in general is scarce. Most research in this field has investigated DWLSs that provide standalone lifestyle therapy, which tend to have a limited impact [23,24]. For example, a 2024 randomised controlled trial of a standalone lifestyle DWLS in Germany reported a mean intervention group weight loss of 3.1% after 24 weeks [23]. Other studies have assessed weight-loss outcomes from standalone GLP-1 RA interventions [25,26]; GLP-1 RA-supported interventions on type 2 diabetes cohorts [27,28]; and GLP-1-RA supported interventions that have been publicly funded [15,29]. Although government subsidisation should be a main objective of DWLSs and their advocates, given its impact on financial access, most DWLSs could remain unsubsidised for the foreseeable future. The only unsubsidised GLP-1 RA-supported DWLS for non-diabetic patients that appears to have been studied on multiple metrics is Juniper—a service that has treated PWOO across Australia, Japan, Germany and the UK since 2021. Studies of the Juniper DWLS have assessed the effectiveness of the service [20,30], patient adherence [13,31], patient satisfaction [32] and patient reasons for using the service [10]. However, all these studies have been confined to Australian and British cohorts.

Digital healthcare uptake in Germany has been increasing rapidly since the COVID-19 pandemic and has been supported by several legislative reforms, including mandates for central patient records and electronic prescriptions [33]. These developments have coincided with the launch of Semaglutide in Germany [34], which has been supplied by several DWLSs, including Juniper. However, while Germany represents Semaglutide's largest European market and could feasibly play a significant role in shifting the nation's

upward overweight and obesity trajectory [34], research on German Semaglutide-supported DWLSs has not been forthcoming.

This study aims to assess the effectiveness of the Juniper DWLS in a cohort of real-world German patients who are reasonably adherent to the program. The Juniper Germany DWLS differs from its Australia and UK equivalents in that its default health coaching is standardised, nutrition focused and delivered via email rather than through a mobile app. The program will be described in detail in the following section. It is believed this study will make a significant contribution to the scarce literature on comprehensive real-world DWLSs by demonstrating the extent to which a previously studied program delivers comparable weight-loss outcomes in a different continent despite differences in lifestyle coaching design and delivery. Outcomes will also be compared with those from a recent study of the fully subsidised Oviva ADHOC standalone lifestyle DWLS in Germany to assess the degree to which GLP-1 RAs can enhance DWLS effectiveness [23].

## 2. Materials and Methods

### 2.1. Study Design

A retrospective cohort study was adopted to achieve the study aims. The investigators followed the Strengthening the Reporting of Observational Studies in Epidemiology (STROBE) guidelines throughout each phase of the study. The Bellberry Limited Human Ethics Committee approved the study on 22 November 2023 (No. 2023-05-563-A-1).

### 2.2. Program Overview

The Juniper Germany DWLS is delivered asynchronously through a web-based platform. Prospective patients complete an online pre-consultation questionnaire, which contains over 100 questions about their health background. Certified doctors assess questionnaire responses and any other requested data, such as blood test results, medical imagery and photos, to determine patient eligibility for the Juniper DWLS. Decisions are based on the European Medicines Agency guidelines for Semaglutide [35], which include (body mass index) BMI cut-off points, contraindications and drug–drug interactions. The BMI cutoffs are 27 kg/m$^2$ for people who have at least one weight-related health condition (e.g., high blood pressure, obstructive sleep apnea, symptomatic cardiovascular disease) and 30 kg/m$^2$ for everyone else. Contraindications include pancreatitis; hypoglycemia and concomitant insulin use; diabetic retinopathy complications; a previous acute kidney injury; multiple endocrine neoplasia syndrome type 2; a family history of medullary thyroid carcinoma; or known hypersensitivity to Semaglutide or any of the product components. Doctors use their discretion in determining whether the medication can be used concomitantly with other oral medications, which may interact with Semaglutide's gastric emptying effect. Once eligible Juniper Germany patients pay their first monthly subscription fee of EUR 356 (same price for each month thereafter), they are allocated an MDT,consisting of a doctor, a university-qualified nutritionist (hereafter referred to as a health coach) and a nurse practitioner. All patients are instructed to use the same set of scales for each weight measurement provided throughout their care journey.

MDTs provide reactive Semaglutide guidance and health coaching via email. At program initiation, health coaches forward patients standardised nutritional recommendations and invite them to ask questions at any stage of their care journey. These recommendations are organised into educational modules under the following topics: caloric deficit, healthy shopping, portion control, macronutrients, protein, adequate water intake, meal guidelines and snack recommendations. Each module is between one to two A4 pages long. Patients can also opt in to receive a detailed meal plan and exercise program in which case they are asked to complete a lifestyle quiz to facilitate program personalisation. When patients ask any questions related to mindset, exercise, or their social life, health coaches provide personalised responses and a link to educational modules on the relevant topics, such as "how to navigate ups and downs", "adding movement to your routine" and "social triggers". However, after program initiation, health coaches do not proactively email patients with

advice at any stage of a patient's care journey, except to confirm a follow-up consultation, which occur every 5 months and on the basis of an ad hoc patient or coach request. Patient subscriptions are cancelled if they fail to schedule follow-up consultations within 20 days after the start of each 5-month interval or from the date of an MDT request for an ad hoc consultation. All follow-up consultations consist of asynchronous questionnaires that solicit information on patient weight, comfort levels, program satisfaction, behavioural changes and anything else the patient would like to share. Responses are reviewed by prescribing doctors, who use their discretion to determine subsequent action. Health coaches access questionnaire responses via patient profiles and provide personalised feedback whenever they consider it necessary. Patients are required to provide weight data at every 5-month follow-up consultation.

To supplement their diet and exercise therapy, Juniper patients are sent a box of GLP-1 RA medications every month. Semaglutide (Ozempic) was the only GLP-1 RA medication prescribed to Juniper Germany patients during the study period. Patients received two reminder emails in the lead-up to each Semaglutide order, informing them that payment will be taken from their linked account unless they cancel their subscription.

Patients are instructed to report side effects whenever they manifest by emailing their MDT with a simple description of the event and giving a severity rating (mild, moderate or severe). In cases of moderate side effects, nurses will schedule ad hoc consults with a patient's doctor. Severe adverse events are immediately referred to emergency services. All reported side effects trigger an alarm in the Eucalyptus clinical auditing system, which allows the service's clinical auditors to escalate the matter appropriately and minimise the risk of a delayed MDT response.

All communication between patients and their MDT is automatically uploaded to Eucalyptus' (Juniper parent company) central data repository on Metabase, including all questionnaire responses and shared multimedia files. To maximise care coordination, each MDT member has complete access to their patients' communications and health profile. All data in the Eucalyptus data repository are encrypted and can only be accessed by MDTs, the Eucalyptus data analytics team and the Eucalyptus clinical auditing team.

### 2.3. Participants

The study included all Juniper Germany patients who subscribed to the DWLS between 28 April 2023 and 28 April 2024, received at least 6 orders of Semaglutide, and submitted weight data between 140 and 170 days after program initiation. These latter two criteria mirror those used in the 2024 effectiveness analysis of the Juniper UK DWLS and were thus replicated to enable a direct comparison with that study.

### 2.4. Measures

The study's coprimary endpoints included mean follow-up weight-loss percentage relative to baseline and the proportion of patients who achieved 5, 10 and 15 percent weight-loss milestones. Side effect incidence and the relationship between demographic characteristics and weight-loss outcomes represented the study's secondary endpoints.

### 2.5. Statistical Analyses

Pearson correlation tests were used to measure the effect of continuous independent variables on weight loss, including age, BMI and days from program initiation to follow-up weight measure. Two-sample *t*-tests were conducted to assess the impact of two-level categorical variables on weight loss, such as gender. These latter tests were substituted by analyses of variance (ANOVA) when independent categorical variables contained three or more levels, such as BMI categories. All analyses were conducted on R Studio (version 2023.06.1+524).

### 3. Results

A total of 2376 patients subscribed to the Juniper Germany DWLS within the study period. Of these patients, 1245 were excluded as a result of receiving less than six Semaglutide orders, 395 of whom were still active patients of the program (i.e., patients who had not been subscribed to the program for long enough to have received six orders). A further 298 patients had received six orders but had failed to submit follow-up weight data within the specified 140–170-day period and were thus also excluded. Consequently, a total of 833 patients were included in the final analysis (Figure 1). Over three quarters (81.87%) of patients in the final analysis were of Caucasian ethnicity and 83.07% were female (Table 1). The mean age at program commencement was 44.85 (±9.81) years and the mean BMI was 34.36 (±5.48) kg/m$^2$.

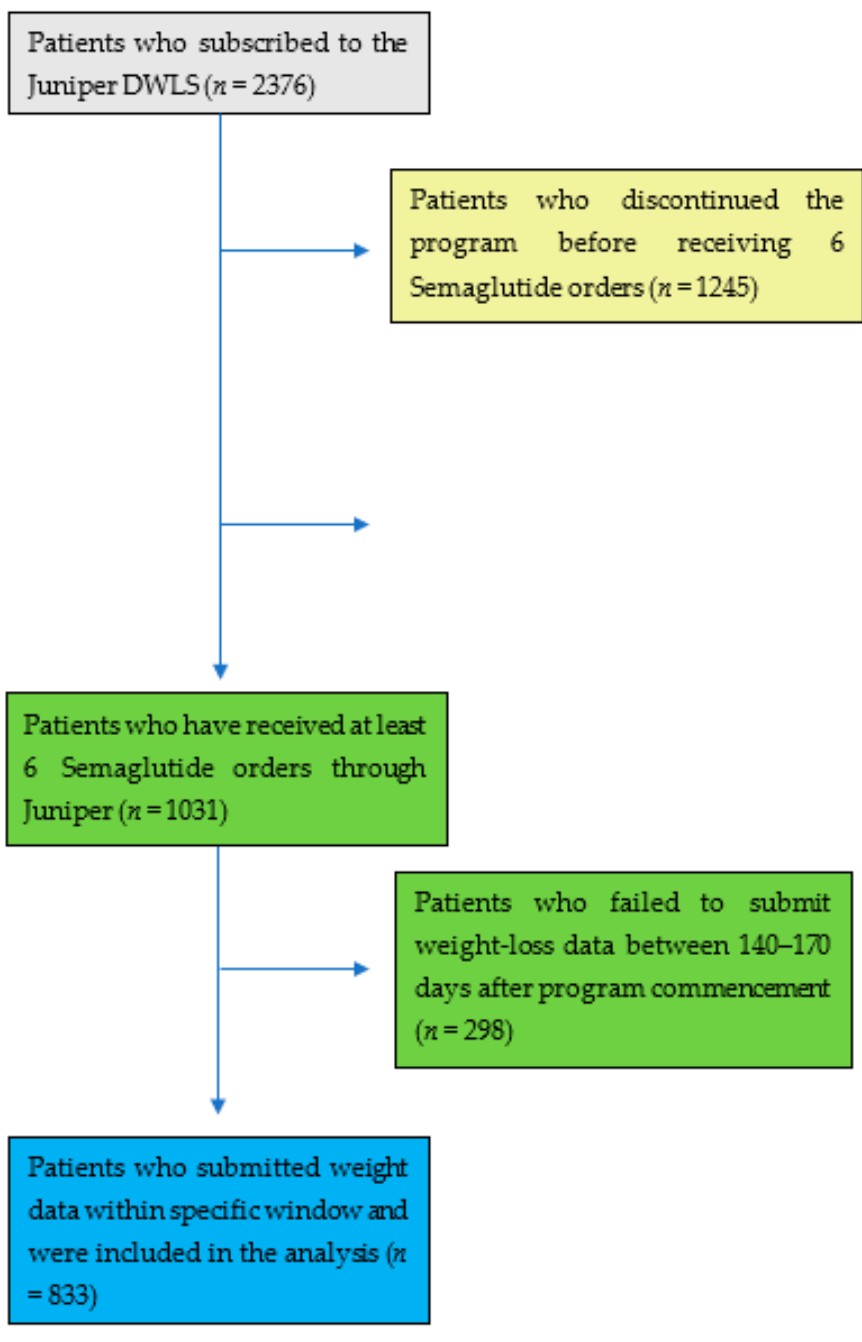

**Figure 1.** Patient Flowchart.

**Table 1.** Baseline characteristics.

| Demographic Information | Mean (SD) |
|---|---|
| Age | 44.85 ($\pm$9.81) years |
| **Gender** | **Number (%)** |
| Female | 692 (83.07) |
| Male | 141 (16.93) |
| **Ethnicity** | **Number (%)** |
| Caucasian | 682 (81.87) |
| Middle Eastern | 63 (7.56) |
| Asian including subcontinent | 41 (4.92) |
| Black African of African Caribbean | 16 (1.92) |
| Latino/Hispanic | 16 (1.92) |
| Rather not say | 15 (1.80) |
| **Clinical information** | **Mean (SD)** |
| BMI | 34.36 ($\pm$5.48) kg/m$^2$ |
| Weight | 99.41 ($\pm$19.66) kg |

Follow-up weight was taken at a mean of 160.14 ($\pm$8.19) days after program initiation. At this point, mean weight loss for the cohort was 9.52 ($\pm$5.46) percent. In terms of weight-loss milestones, 81.51% of the cohort lost 5% or more of their baseline weight; 45.14% lost at least 10%; and 14.17% of patients achieved the 15% milestone.

A two-sample *t*-test revealed that the mean weight loss percentage was statistically higher among female patients (*Mean* = 9.75) relative to male patients (*Mean* = 8.41), $t(831) = 2.67$, $p < 0.01$ (Table 2). Due to the low representation of non-Caucasian ethnic groups, we merged each of these categories to create a binary ethnicity variable. A subsequent two-sample *t*-test showed that weight-loss percentage did not correlate with ethnicity, $t(831) = 0.04$, $p = 0.97$. Pearson tests found a small but statistically significant positive association between weight loss percentage and days-to-follow-up-weight-measure, $r(831) = 0.11$, $p = 0.001$; a small, but statistically significant negative relationship between weight loss percentage and BMI, $r(831) = -0.15$, $p < 0.001$; and no significant correlation between age and weight loss percentage, $r(831) = -0.04$, $p = 0.229$ (Table 3). To gain a deeper understanding of the association between initial BMI and weight loss percentage, four BMI categories were created and a subsequent ANOVA was conducted. The analysis revealed a statistically significant association between the variables, ($F(3830) = 3.82$, $p < 0.01$), and a Tukey post hoc test showed that association stemmed from differences between Class 3 obesity (BMI > 40 kg/m$^2$; *Mean* weight loss = 8.11%) and both Class 1 obesity (BMI 30–35 kg/m$^2$; *Mean* weight loss = 9.74%) and the overweight category (BMI 27.5–29.99 kg/m$^2$; *M* weight loss = 10.1) (Table 4) (Figure 2). Similarly, an ANOVA ($F(3830) = 4.83$, $p < 0.01$) and Tukey post hoc test found that patients who submitted follow-up weight data between 160–170 days after program initiation lost significantly more weight (*Mean* = 9.98%) than those from the 140–149.99- (*Mean* = 9%) and 150–159.99-day (*Mean* = 8.61%) categories (Figure 3).

**Table 2.** Weight loss by gender and ethnicity.

| | Female | | Male | | | | | |
|---|---|---|---|---|---|---|---|---|
| | **Mean** | **SD** | **Mean** | **SD** | *df* | *t* | *p* | **Cohen's d** |
| Weight loss | 9.75 | 5.51 | 8.41 | 5.12 | 831 | 2.67 | <0.01 | |
| | **Caucasian** | | **Non-Caucasian** | | | | | |
| | **Mean** | **SD** | **Mean** | **SD** | *df* | *t* | *p* | **Cohen's d** |
| Weight loss | 9.53 | 5.44 | 9.51 | 5.54 | 831 | −0.04 | 0.97 | |

**Table 3.** Pearson correlations between weight loss and continuous independent variables.

|  | Weight Loss (%) | Age (years) | BMI (kg/m$^2$) | Days to Weight Measure |
|---|---|---|---|---|
| Weight loss (%) | 1.00 | −0.04 | −0.15 *** | 0.11 ** |
| Age (years | −0.04 | 1.00 | −0.02 | 0.02 |
| BMI (kg/m$^2$) | −0.15 *** | −0.03 | 1.00 | −0.01 |
| Days to weight measure | 0.11 ** | 0.02 | −0.01 | 1.00 |

Note: *** $p < 0.001$, ** $p < 0.01$.

**Table 4.** Post Hoc Tukey test results.

| Weight Loss (%) | BMI Categories | Levels | Mean Difference | *p* Value |
|---|---|---|---|---|
|  |  | 30–34.99:27.5–29.99 | −0.35 | 0.9 |
|  |  | 35–39.99:27.5–29.99 | −0.47 | 0.86 |
|  |  | 40 and over:27.5–29.99 | −1.98 | <0.01 ** |
|  |  | 35–39.99:30–34.99 | −0.12 | 0.99 |
|  |  | 40 and over:30–34.99 | −1.63 | 0.02 * |
|  |  | 40 and over:35–39.99 | −1.51 | 0.08 |
| Weight loss (%) | Days to FU measure | Levels | Mean difference | *p* value |
|  |  | 150–159.99:140–149.99 | 0.26 | 0.91 |
|  |  | 160–169.99:140–149.99 | 1.36 | 0.04 * |
|  |  | 160–169.99:150–159.99 | 1.1 | 0.03 * |

Note: * $p < 0.05$, ** $p < 0.01$.

Side effects were reported by 60.98% of patients, among which gastrointestinal issues were the most common (49.22% of all reported adverse events), followed by headaches (22.44%) and fatigue or dizziness (19.93%) (Table 5). Over three quarters of all reported side effects were of mild severity (75.81%), with 22.31% considered moderate and 1.88% considered severe. No deaths were reported and no study patients were hospitalised.

**Table 5.** Side effects.

|  | Number (% of Cohort) |  |  |
|---|---|---|---|
| Gastrointestinal issues | 410 (49.22) |  |  |
| Headaches | 187 (22.44) |  |  |
| Fatigue or dizziness | 166 (19.93 |  |  |
| Mood changes | 123 (14.77) |  |  |
| Other | 73 (8.76) |  |  |
| Total patients with at least one side effect | 508 (60.98) |  |  |
|  | **Number (% of total reported side effects)** |  |  |
| Mild side effects | 727 (75.81) |  |  |
| Moderate side effects | 214 (22.31) |  |  |
| Severe side effects | 18 (1.88) |  |  |
| **Side effect type by severity level—number (% of total number of side effect type)** | **Mild** | **Moderate** | **Severe** |
| Gastrointestinal issues | 299 (72.92) | 103 (25.12) | 8 (1.95) |
| Headaches | 114 (60.96) | 67 (35.83) | 6 (3.2) |
| Fatigue or dizziness | 120 (72.29) | 44 (26.51) | 2 (1.2) |
| Mood changes | 108 (87.8) | 13 (10.57) | 2 (1.63) |
| Other | 64 (87.67) | 9 (12.33) | 0 (0) |

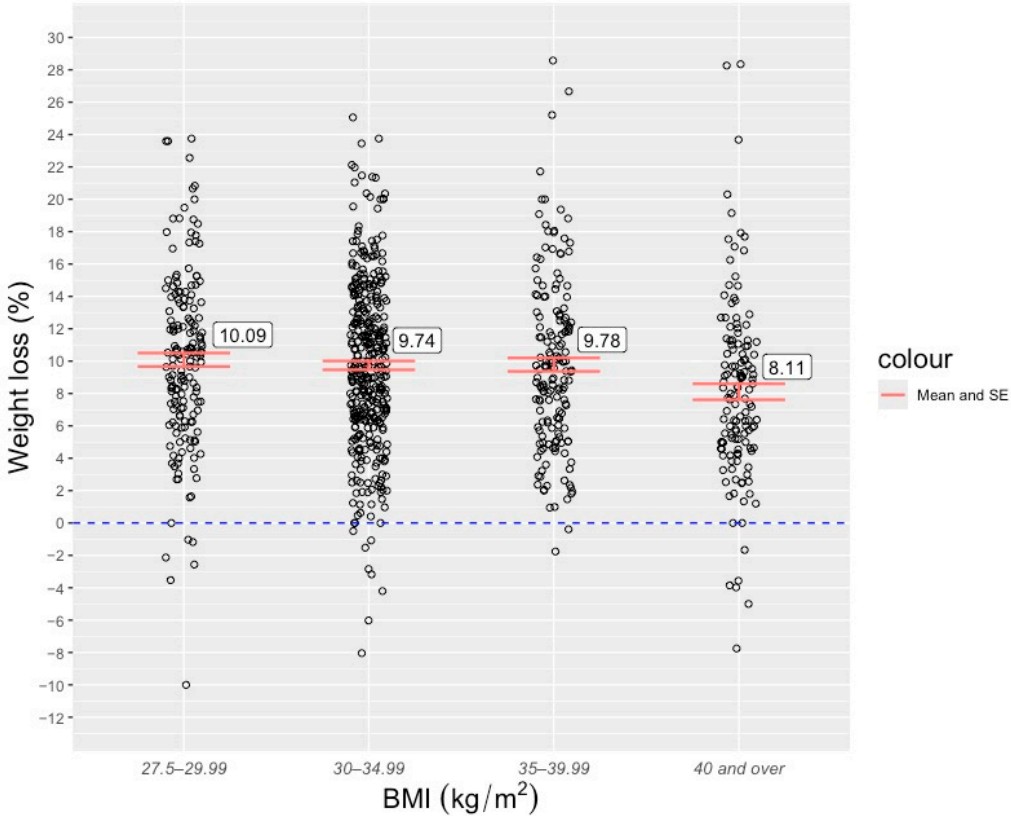

**Figure 2.** Weight-loss percentage by BMI category.

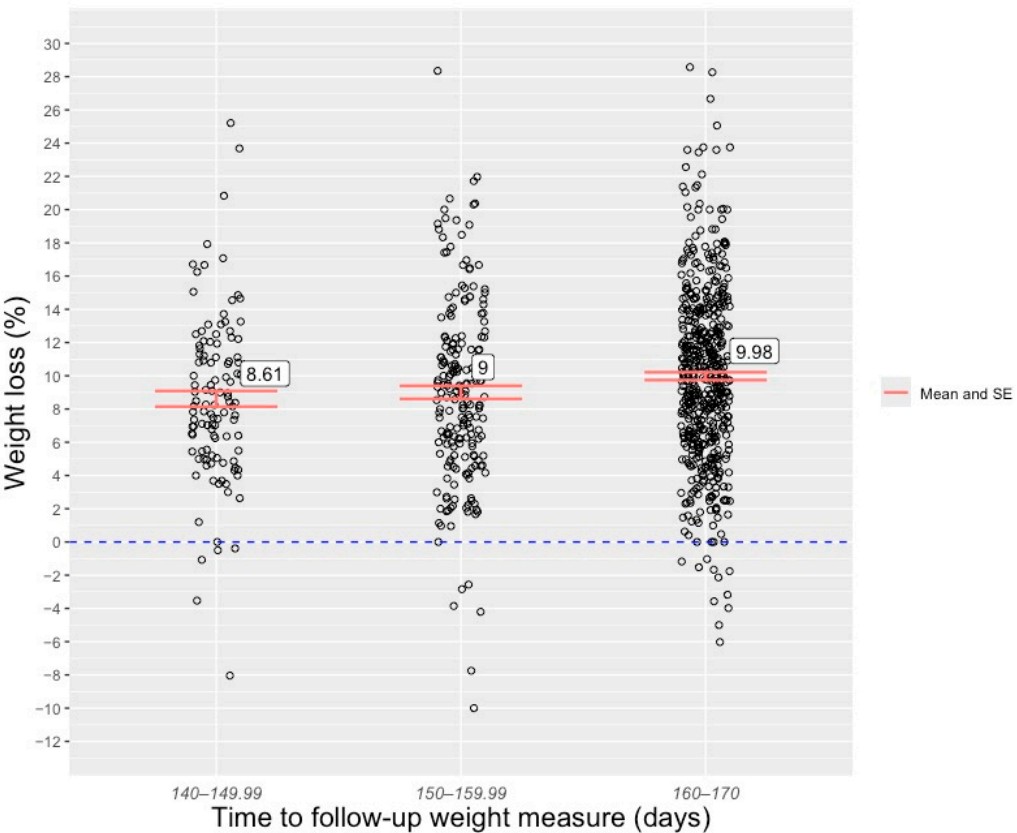

**Figure 3.** Weight loss by days-to-follow-up-weight-measure categories.

## 4. Discussion

This is the first study to investigate weight-loss outcomes from a real-world Semaglutide-supported DWLS in continental Europe. An increasingly large number of PWOO have been accessing GLP-1 RAs such as Semaglutide since clinical trials began to reveal the medications' unprecedented effect on weight loss in the mid-2010s [2,3,36]. Despite these trials consistently revealing comparable safety outcomes across control and intervention groups, esteemed medical institutions such as the WHO and NICE stress that GLP-1 RAs should only ever be used as an adjunct to lifestyle therapy under the guidance of a coordinated MDT [8,9]. However, the effectiveness of such services in real-world settings, where PWOO have to pay high fees and juggle their therapy with significant family and work commitments, is largely unknown. Only two studies have hitherto measured the effectiveness of unsubsidised GLP-1 RA-supported DWLSs [20,30]. These studies took place in Australia and the UK, where lifestyles could feasibly have been different enough from German or general European culture to impact the effectiveness of such services. Germany is also Europe's largest Semaglutide market and has experienced a rapid uptake of digital healthcare since the COVID-19 pandemic. Other points of distinction for this study were the design of its default coaching material and the coaching medium. Whereas British and Australian Juniper cohorts from previous studies all received personalised diet and exercise advice, Juniper Germany patients in this study received standardised nutritional guidance and had to opt in for personalised diet and exercise plans. Juniper Germany patients communicated with MDTs exclusively via email rather than a mobile app, as was the case in the Australian and British studies.

The analysis found that the Juniper Germany patients lost, on average, 9.52% of their baseline weight after 5 months (*Mean* = 160.14 days post initiation). Over four-fifths of the cohort (81.51%) achieved at least five percent weight loss, an amount widely regarded as the benchmark of 'meaningful' weight loss [37]. Both the mean figure and proportion of patients who achieved 'meaningful' weight loss were roughly one percentage point lower than those observed in the Juniper UK study (*Mean* =10.73%; 5% weight loss = 82.36%) (Figure 4).

The mean days-to-follow-up-weight-measure was also lower in the UK study (153.84 vs. 160.14). While a disparity of 6.3 days may not appear significant, the results from the Tukey Post Hoc Test in this Juniper Germany study indicate that increasing the figure by such a period in the UK study could have led to a greater difference in mean weight loss between the two cohorts. The difference in the proportion of patients who achieved meaningful weight loss (81.51% vs. 82.36%) may have also been greater, but the real-world significance of this change would arguably be less pronounced given that both figures are already relatively high. Aside from this disparity in the mean days-to-follow-up-weight-measure, all baseline variables were largely consistent with the Juniper UK study, including the mean age of 44.85 years (versus 45.2 in the UK study) and mean BMI of 34.36 kg/m$^2$ (versus 34.6 in the UK study). As a result of this congruity, we can loosely attribute the relative inferiority of the Juniper Germany cohort's weight-loss outcomes (relative to Juniper UK cohort) to the difference in health coaching design, its medium of delivery and/or various cultural differences.

While the above conclusion may appear weak, its significance should not be under-estimated in the context of global obesity rates and the scarcity of literature on GLP-1 RA-supported DWLSs. Previous research has indicated that comprehensive DWLSs can mitigate considerable access barriers to obesity care [10] and be adhered to for significant periods [13,31]. The other crucial factor in the question as to whether DWLSs can play an important role in shifting alarming obesity trajectories is their effectiveness in real-world settings. Although this study observed inferior outcomes to a near identical study on a UK-based cohort, it still found that the average adherent Juniper Germany patient lost a significant amount of weight after roughly 5 months and that over four-fifths of adherent patients lost a 'meaningful' amount of weight. These findings suggest that comprehensive DWLSs such as Juniper, can be effective in a variety of cultures and through different health coaching mediums. In comparison, the 2024 study of the Oviva digital standalone lifestyle

intervention in Germany reported a mean weight loss of 3.1% after 24 weeks (5.5 months) (Figure 5).

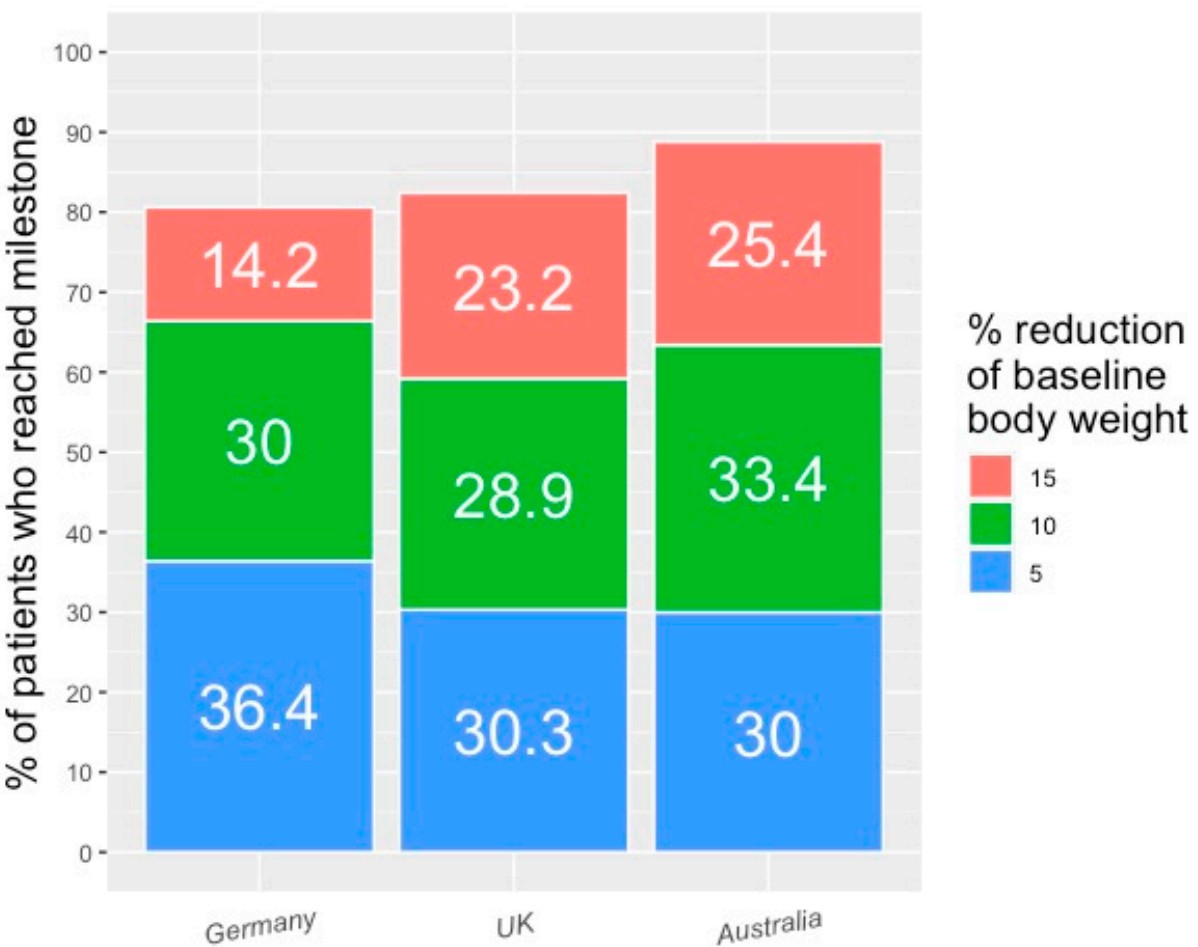

**Figure 4.** Comparison of weight-loss outcomes from Juniper DWLSs in Germany, the UK and Australia. NB: Mean follow-up weight measures were taken 153.84 (±6.66) days and 226.69 (±12.81) after program initiation of the UK and Australia Juniper studies, respectively. Patients in the Australia study had their health coaching supplemented with liraglutide rather than Semaglutide.

While the proportion of patients who discontinued this standalone lifestyle intervention before the 5.5-month point was lower (29.76%) than the figure observed in the Semaglutide-supported Juniper cohort (47.96%), definitive conclusions about drop-out rates can only be drawn from dedicated adherence studies, which measure mean adherence (days) and discontinuation reason distributions. Previous adherence studies of Juniper DWLSs have, for example, discovered that a relatively small proportion of patients discontinued due to side effects (3.8% Australia; 15.4% UK) or service dissatisfaction (7.2% Australia; 2.4% UK) [13,31]. Most patients from these cohorts dropped out for reasons impacted by extrinsic factors, such as GLP-1 RA supply, program cost or personal weight-loss expectations (either met or unmet), which would also feasibly be influenced by program cost. A dedicated adherence analysis of the Juniper Germany DWLS should follow this study. Future research should also aim to further explore this study's finding that patients from lower BMI categories lost significantly more weight than those from higher BMI categories.

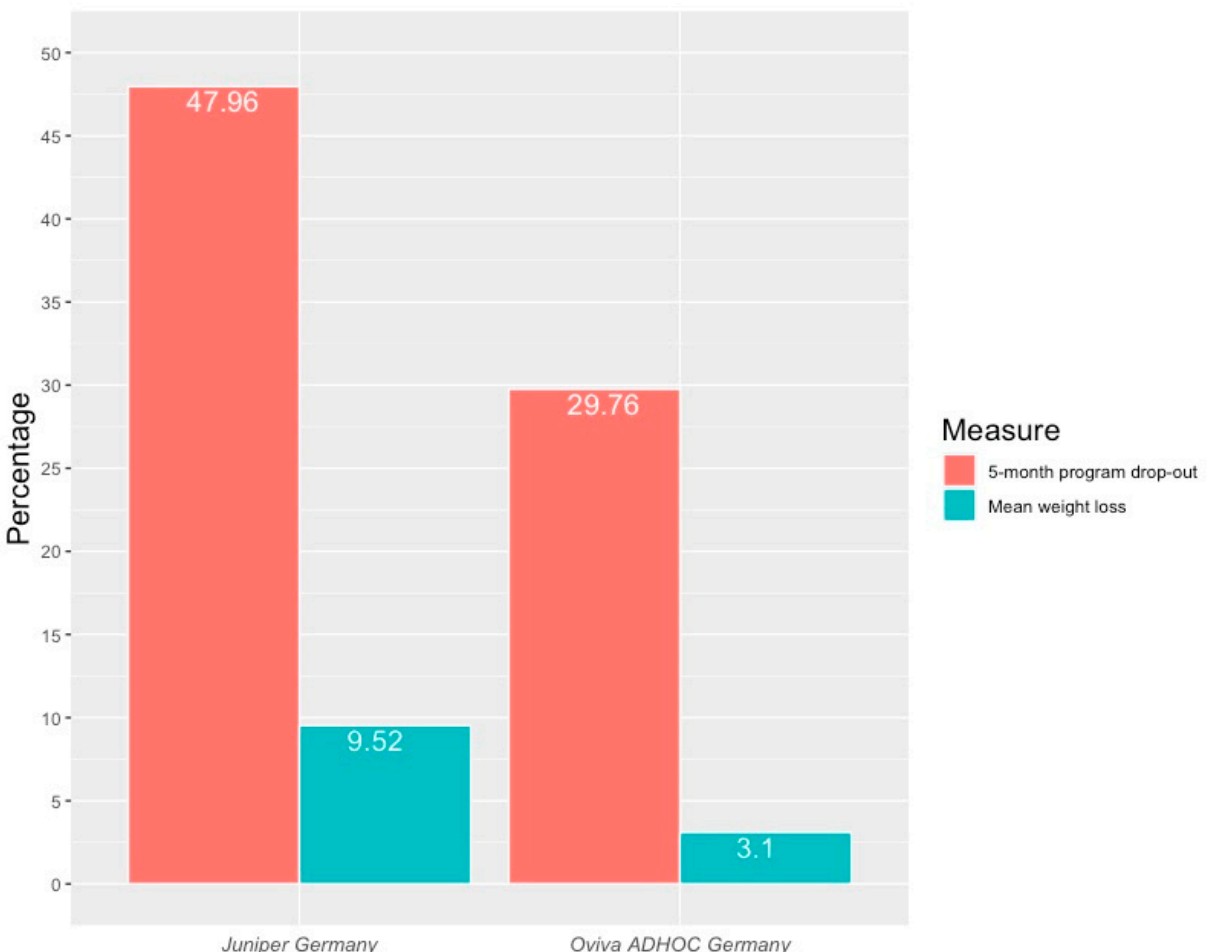

**Figure 5.** Comparison of 5-month weight-loss and drop-out rate between Juniper Germany and Oviva ADHOC Germany intervention. NB: Measurements of the Oviva ADHOC cohort were taken after 24 weeks (168 days/5.5 months), whereas follow-up measurements of the Juniper Germany cohort were taken after a mean of 160.14 days (22.9 weeks/5.3 months). The Juniper drop-out rate equation excluded still active patients ($n$ = 395) from the denominator and patients who did not submit weight data in the specified study period (140–170 days) ($n$ = 298) from the numerator, thus becoming $(1031 - 1981)/1981 \times 100$. The drop-out rate of the Oviva ADHOC group was calculated by dividing the number of participants who completed the 24-week assessment ($n$ = 59) by the total number of participants who received the allocated intervention ($n$ = 84) after initial exclusions.

This study had several limitations. Firstly, the sample contained a disproportionate number of females and was not representative of the ethnic diversity in German society. Secondly, all weight data were patient-reported and therefore may have been subjected to personal biases. Although patients were instructed to use the same set of scales for each weight measurement, it is possible that such advice was not adhered to and that measurements were thus less reliable than those from the Juniper UK and Australia studies in which patients had access to standardised Bluetooth scales. Thirdly, the study assessed outcomes at 5 months and thus only enables an assessment of the short-term effectiveness of the Juniper Germany DWLS. Furthermore, the Eucalyptus analytics team did not systematically identify patients who opted for personalised meal and exercise plans during the study period, so investigators were not able to compare the outcomes of such patients with those who only received standardised guidance. Such comparisons should be made in future research. Finally, the investigators could only report weight-loss outcomes on 35.06% of patients who subscribed to the Juniper Germany DWLS within the study period, given the study's significant inclusion criteria. Investigators could explain that 395 (16.62%)

active patients were excluded because they had not been on the program long enough to have received six Semaglutide orders, and a further 298 (12.54%) patients failed to submit weight data between 140 and 170 days. However, they were not able to account for the precise reason the remaining 850 patients discontinued the Juniper Germany DWLS before their sixth order of Semaglutide. Investigators could not report weight-loss outcomes of early program discontinuers, as patients were only required to provide weight data at the 5-month follow-up consultation. Consequently, the weight-loss findings reported in this study only reflect the outcomes of reasonably adherent patients. A dedicated adherence study similar to those that have been conducted on the Juniper Australia and UK cohorts is needed to accurately report the Juniper Germany DWLS attrition rate.

## 5. Conclusions

Previous research has demonstrated the potential of GLP-1 RA-supported DWLSs as an effective weight-loss intervention for PWOO in Australia and the UK. The findings from this study indicate that such services can also deliver good outcomes for German patients who reasonably adhere to program protocol, observing a mean weight loss of 9.52% after 5 months (*Mean* = 160.14 days). These outcomes compared favourably to the 3.1% mean weight loss reported earlier this year in a 24-week (168 days) study of a standalone lifestyle DWLS in Germany [23]. Moreover, over four-fifths of adherent Juniper patients lost what the literature commonly considers the benchmark for a meaningful amount of weight (at least 5%) [37], which few patients from standalone lifestyle DWLSs tend to achieve. Future research should aim to conduct a dedicated adherence analysis of the Juniper Germany DWLS and investigate the impact of baseline BMI and subsidisation on general DWLS effectiveness. Overall, this study adds an important foundational layer to the scarce literature on real-world GLP-1 RA-supported DWLSs.

**Author Contributions:** Conceptualisation, L.T. and M.V.; methodology, L.T., L.R. and M.V.; validation, L.T., L.R. and M.V.; formal analysis, L.T. and M.V.; investigation, L.T.; resources, M.V. and L.R.; data curation, L.T. and L.R.; writing—original draft preparation, L.T.; writing—review and editing, L.T. and M.V.; supervision, L.T.; project administration, L.T.; visualisation, L.T.; software, L.T. All authors have read and agreed to the published version of the manuscript.

**Funding:** This research received no external funding.

**Institutional Review Board Statement:** This study was conducted in accordance with the Declaration of Helsinki and approved by the institutional review board (or ethics committee) of the Bellberry Ethics Committee (No. 2023-05-563-A-1, 22 November 2023).

**Informed Consent Statement:** Eucalyptus patients consented to the service's privacy policy at subscription, which includes permission to use their de-identified data for research.

**Data Availability Statement:** The data presented in this study are available from the corresponding author on reasonable request.

**Acknowledgments:** The authors would like to thank all patients, clinicians and auditors involved in the Eucalyptus weight loss program over the study period.

**Conflicts of Interest:** L.T., M.V. and L.R. are paid a salary by Eucalyptus.

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
