# Peer review of "Effectiveness of an Email-Based, Semaglutide-Supported Weight-Loss Service for People with Overweight and Obesity in Germany: A Real-World Retrospective Cohort Analysis"

_2673-4168, doi:10.3390/obesities4030021_

Round 1

Reviewer 1 Report

Comments and Suggestions for Authors

Author Response

Comment:

The study also compares the service's outcomes in Germany, the UK, and Australia. To further improve the impact of this study, the efficacy of Semaglutide-supported digital weight loss service should be benchmarked against other weight loss services. The inability to compare with other weight loss services leaves unanswered questions about the relative effectiveness of Semaglutide-supported digital weight loss services within the broader spectrum of available weight loss services options. Including such comparisons would strengthen the study by providing context and highlighting the unique benefits or limitations of the service evaluated.

Response:

Thank you for this vital recommendation. We have now included a standalone lifestyle DWLS comparator to the study, which measured outcomes over nearly the exact same period in a German cohort of patients. We present this comparator in the introduction in the following sentences:

Lines 88-92 – “Most research in this field has investigated DWLSs that provide standalone lifestyle therapy, which tend to have a limited impact [23][24]. For example, a 2024 randomized controlled trial of a standalone lifestyle DWLS in Germany reported mean intervention group weight loss of 3.1% after 24 weeks [23].”

Lines 122-124 – “Outcomes will also be compared with those from a recent study of the fully subsidized Oviva ADHOC standalone lifestyle DWLS in Germany to assess the degree to which GLP-1 RAs can enhance DWLS effectiveness [23].”     

We compare outcomes in the discussion section in a bar chart (Figure 4) and the following sentences:

Lines 532-557 –“In comparison, the study of Oviva digital standalone lifestyle intervention in Germany reported mean weight loss of 3.1% after 24 weeks (5.5 months) (Figure 4). And while the proportion of patients who discontinued this standalone lifestyle intervention before the 5.5-month point was lower (29.76%) than the figure observed in the Semaglutide-supported Juniper cohort (47.96%), definitive conclusions about drop-out rates can only be drawn from dedicated adherence studies, which measure mean adherence (days) and discontinuation reason distributions. Previous adherence studies of Juniper DWLSs have, for example, discovered that a relatively small proportion of patients discontinued due to side effects (3.8% Australia; 15.4% UK) or service dissatisfaction (7.2% Australia; 2.4% UK) [13][31]. Most patients from these cohorts dropped out for reasons impacted by extrinsic factors, such as GLP-1 RA supply and program cost; or personal weight-loss expectations (either met or unmet), which would also feasibly be influenced by program cost. A dedicated adherence analysis of the Juniper Germany DWLS should follow this study.”

We have also now summarized the comparison in the conclusion:

Lines 593-598 – “These outcomes compared favorably to the 3.1% mean weight loss reported earlier this year in a 24-week (168 days) study of a standalone lifestyle DWLS in Germany [23]. Moreover, over four-fifths of adherent Juniper patients lost what the literature commonly considers the benchmark for a meaningful amount of weight (at least 5%) [36], which few patients from standalone lifestyle DWLSs tend to achieve.”

Comment:

Furthermore, the inability to compare the completers vs non-completers data also undermines our ability to understand the effectiveness of the intervention. In real-life, patients dropped out for various reasons, do they dropped out because they’ve lost less weight or whether they’ve achieved desired weight loss?

Response:

Thank you for raising this important question. We stress throughout the study that the effectiveness measures are only taken from ‘adherent patients/patients who reasonably adhere to program protocol’. We list this as one of the study’s limitations and have now added a sentence (lines 580-582) to clarify that ‘non-completer’ outcomes could not be reported as Juniper patients were only required to provide weight data at the 5-month follow-up consultation:

“Investigators could not report weight-loss outcomes of early program discontinuers, as patients were only required to provide weight data at the 5-month follow-up consultation.”

We highlight in the limitations paragraph that we “were not able to account for the precise reason the remaining 850 patients discontinued the Juniper Germany DWLS before their sixth order of Semaglutide” (lines 578-580). However, we have now indicated that a significant proportion of patients likely dropped-out for reasons unrelated to side effects or service dissatisfaction by summarizing outcomes from dedicated adherence studies of the Juniper Aus and UK DWLSs and stressing that a similar study of the German service should be conducted:

Lines 548-557 - “…definitive conclusions about drop-out rates can only be drawn from dedicated adherence studies, which measure mean adherence (days) and discontinuation reason distributions. Previous adherence studies of Juniper DWLSs have, for example, discovered that a relatively small proportion of patients discontinued due to side effects (3.8% Australia; 15.4% UK) or service dissatisfaction (7.2% Australia; 2.4% UK) [13][31]. Most patients from these cohorts dropped out for reasons impacted by extrinsic factors, such as GLP-1 RA supply and program cost; or personal weight-loss expectations (either met or unmet), which would also feasibly be influenced by program cost. A dedicated adherence analysis of the Juniper Germany DWLS should follow this study.”

Once again, thank you for raising these vital questions about the program’s adherence rate, which we totally support. However, we believe that conducting an effectiveness and adherence analysis together in the same retrospective study would render its scope too large. A dedicated adherence study on the Juniper Germany program will be our next priority.

Comment:

Throughout the manuscript, Is M = mean or median? I do not recommend abbreviate mean or median. Standard deviation is required when reported as mean, whereas interquartile range is required when reported as median.

Response:

Thank you for this comment. We have now replaced all ‘M =’ abbreviations with “Mean =”.

Comment:

  1. Line 20 – 21: I recommend specify the BMI categories.

Response:

Thank you for this comment. We have now specified the BMI categories in the relevant parentheses.

Comment:

  1. Lines 34 – 35: It’s beter avoiding the phrase “address the genetic component of weight management”, it may sound like genetic modification.

Response:

Thank you for this important observation. We have now changed the phrase to “…they treat the neurological component of weight management by…”

Comment:

  1. Line 45: I’m struggling to understand what is MDT? The abbreviation has not been expanded elsewhere.

Response:

Thank you for noticing this. We have now expanded the acronym to ‘Multidisciplinary teams’ at its first mention on line 50.

Comment:

  1. Line 61, “This later feature”: Please specify what is the feature.

Response:

Thank you for picking up on this vague expression. We have now changed it to “This asynchronous feature” .

Comment:

  1. Line 64: OECD abbreviation has not been expanded elsewhere.

Response:

Thank you for noticing this. We have now expanded the acronym.

Comment:

  1. Line 89, “a service that has treated over 80,000 PWOO across Australia, Japan, Germany and the UK since 2021”: This sounds too much like marketing than science. It is questionable whether the 80,000 people are really “treated”? How is “treated” defined in this context?

Response:

Thank you for making this important point. We have now removed the ’over 80,000’ figure from the phrase.

Comment:

  1. Line 121 – 123: Please state the inclusion and exclusion criteria for the study.

Response:

Thank you for requesting these important details. We have now added the following information to the program overview section (line140-154)

“The BMI cutoffs are 27kg/m2 for people who have at least one weight-related health condition (e.g., high blood pressure, obstructive sleep apnea, symptomatic cardiovascular disease), and 30kg/m2 for everyone else. Contraindications include pancreatitis; hypoglycemia and concomitant insulin use; diabetic retinopathy complications; a previous acute kidney injury; multiple endocrine neoplasia syndrome type 2; a family history of medullary thyroid carcinoma; or known hypersensitivity to Semaglutide or any of the product components. Doctors use their discretion in determining whether the medication can be used concomitantly with other oral medications, which may interact with Semaglutide’s gastric emptying effect.”

Comment:

  1. Line 127: Please state the model, brand, and manufacture location of the weighing scale. Please also describe the instructions given to the patients, for example the frequency of weighing. Is the electronic scale automatically transmiƫting data to the clinic?

Response:

Thank you for this important request. We misinterpreted the detailed program description that was sent to us from the Juniper coaching team, which contained a footnote that explained that three features (relevant to this study) have only come into effect recently (i.e., after the study period). The sentence detailing the use of scales (line 157-159) now reads like this:

“All patients are instructed to use the same set of scales for each weight measurement provided throughout their care journey.”

We have now also referred to this as a further limitation in the discussion section (line 564-567), as follows:

“Although patients were instructed to use the same set of scales for each weight measurement, it is possible that such advice was not adhered to and that measurements were thus less reliable than those from Juniper UK and Australia studies in which patients had access to standardized Bluetooth scales.”  

Comment:

  1. How is line 129 - 130 differs from line 130 - 131? Detailed information on the diet and exercise plan is much appreciated so that others can replicate the success of this intervention, or improve on the limitations of this intervention.

Response:

Thank you for identifying the lack of detail here. As above, this aspect of the program was slightly misinterpreted due to our failure to read the footnote information, which stated that the program had changed since the study period. We have now removed the tautological sentence and added specific details about the standardized and opt-in features. The first half of the paragraph (Line 160-171) now reads like this:

‘MDTs provide reactive Semaglutide guidance and health coaching via email. At program initiation, health coaches forward patients standardized nutritional recommendations and invite them to ask questions at any stage of their care journey. These recommendations are organized into educational modules under the following topics: caloric deficit, healthy shopping, portion control, macronutrients, protein, adequate water intake, meal guidelines, and snack recommendations. Each module is between one to two pages long. Patients can also opt in to receive a detailed meal plan and exercise program, in which case they are asked to complete a lifestyle quiz to facilitate program personalization. When patients ask any questions related to mindset, exercise, or their social life, health coaches provide personalized responses and a link to educational modules on the relevant topics, such as “how to navigate ups and downs”, “adding movement to your routine”, and “social triggers”. ‘

 We also made slight changes to a sentences in the introduction (Lines 116-117) and discussion (lines 473-477) to reflect these updated details:

Intro- “The Juniper Germany DWLS differs from its Australia and UK equivalents in that its default health coaching is standardized and nutrition focused, and is delivered via email rather than through a mobile app.”

Discussion Other points of distinction for this study were the design of its default coaching material and the coaching medium. Whereas British and Australian Juniper cohorts from previous studies all received personalized diet and exercise advice, Juniper Germany patients in this study received standardized nutritional guidance and had to opt in for personalized diet and exercise plans.”

Finally, we acknowledged the limitation of these updated details by adding a sentence to the limitations paragraph the end of the discussion section (Lines 569-573):

“Furthermore, the Eucalyptus analytics team did not systematically identify patients who opted for personalized meal and exercise plans during the study period, so investigators were not able to compare the outcomes of such patients with those who only received standardized guidance. Such comparisons should be made in future research.”

Comment:

  1. Line 136: What happens during the consultation (especially during Month 3 and 5, whereby the patients are still in the study)? Is the consultation by texts, telephone calls, video calls, or in-person interview?

Response:

Thank you for requesting these important details. This was the third and final detail missed in the footnote of the program information sheet. During the study period, patients had compulsory 5-month follow-up consultations and ad-hoc consultations at the request of their coach or the patient themselves. These details and the mode of consultation delivery are now captured in the revised lines (171-183):

“However, after program initiation, health coaches do not proactively email patients with advice at any stage of a patient’s care journey, except to confirm a follow-up consultation, which occur every 5 months and on the basis of an ad-hoc patient or coach request. Patient subscriptions are cancelled if they fail to schedule follow-up consultations within 20 days after the start of each 5-month interval or from the date of an MDT request for an ad-hoc consultation. All follow-up consultations consist of asynchronous questionnaires that solicit information on patient weight, comfort levels, program satisfaction, behavioural changes and anything else the patient would like to share. Responses are reviewed by prescribing doctors who use their discretion to determine subsequent action. Health coaches access questionnaire responses via patient profiles and provide personalized feedback whenever they consider it necessary. Patients are required to provide weight data at every 5-month follow-up consultation.”

Comment:

  1. Line 179 – 184: Do the authors consider using a flow diagram?

Response:

Thank you for this excellent suggestion. A flow chart has now been included as Figure 1.

Comment:

  1. Line 179 – 184: 1245 participants were exlcuded due to receiving less than 6 Semaglutide orders. Do they fail to adhere to the intervention? Is the weight data available for these participants? Whilst it is important to analyse the completers data, it is equally crucial to compare the weight loss in those who fully adhere vs those who fail to adhere. Do they dropped out because they’ve lost less weight or whether they’ve achieved desired weight loss? Do the authors have monthly body weight data for these participants?

Response:

Thank you for raising these key questions. We state in the limitations section that “395 (16.62%) active patients were excluded because they had not been on the program long enough to have received 6 Semaglutide orders” (lines 575-577) and that we “were not able to account for the precise reason the remaining 850 patients discontinued the Juniper Germany DWLS before their sixth order of Semaglutide” (lines 578-580). We have now added a sentence (lines 580-582) to clarify that ‘non-completer’ outcomes could not be reported as Juniper patients were only required to provide weight data at the 5-month follow-up consultation:

“Investigators could not report weight-loss outcomes of early program discontinuers, as patients were only required to provide weight data at the 5-month follow-up consultation.”

Consequently, we stress at the end of the discussion (lines 582-587) that:

“the weight-loss findings reported in this study only reflect the outcomes of reasonably adherent patients. A dedicated adherence study is needed to accurately report the Juniper Germany DWLS attrition rate, such as those that have been conducted on Juniper Australia and UK cohorts.”

We have also now indicated that a significant proportion of patients likely dropped-out for reasons unrelated to side effects or service dissatisfaction by summarizing outcomes from dedicated adherence studies of the Juniper Aus and UK DWLSs and emphasizing that a similar study of the German service should be conducted:

Lines 548-557 - “…definitive conclusions about drop-out rates can only be drawn from dedicated adherence studies, which measure mean adherence (days) and discontinuation reason distributions. Previous adherence studies of Juniper DWLSs have, for example, discovered that a relatively small proportion of patients discontinued due to side effects (3.8% Australia; 15.4% UK) or service dissatisfaction (7.2% Australia; 2.4% UK) [13][31]. Most patients from these cohorts dropped out for reasons impacted by extrinsic factors, such as GLP-1 RA supply and program cost; or personal weight-loss expectations (either met or unmet), which would also feasibly be influenced by program cost. A dedicated adherence analysis of the Juniper Germany DWLS should follow this study.”

Once again, thank you for raising these vital questions about the program’s adherence rate, which we totally support. However, we believe that conducting an effectiveness and adherence analysis together in the same retrospective study would render its scope too large. A dedicated adherence study on the Juniper Germany program will be our next priority.

Comment:

  1. Figure 1: Comparison with other studies should be in Discussion, not Results.

Response:

Thank you for noticing this error. We have now moved this figure (Figure 4 in revised manuscript) to the discussion section.

Comment:

  1. Line 220: Describe the difference in weight loss by days to follow-up in text, similar to how the authors described difference in weight loss by BMI category in text.

Response:

Thank you for noticing this inconsistency. We have now added the requested details to the sentence (line 403-407) so that it reads like this:

“Similarly, an ANOVA (F(3,830) = 4.83, p < 0.01) and Tukey post hoc test found that patients who submitted follow-up weight data between 160-170 days after program initiation lost significantly more weight (Mean = 9.98) than those from the 140-149.99- (Mean = 9) and 150-159.99-day (Mean = 8.61) categories (Figure 3).”

Comment:

  1. Table 4: Not required.

Response:

Thank you for this suggestion. We have now removed table 4.

Comment:

  1. Table 5: This information can be presented in Figure 2.

Response:

Thank you for this suggestion. Although we see the benefit of adding as much detail to figures as possible, we believe that including Post Hoc Tukey results in Figure 2 (now Figure 3) results will render it too messy and incomprehensible. We are however happy to make this change if you feel it is absolutely necessary.

Comment:

  1. Figure 2 and 3: Units are required on both axes.

Response:

Thank you for noticing the absences of units of the axis titles. These have now been added to the referred Figures (Now Figures 2 and 3).   

Comment:

  1. Table 6: Can the authors quantify the severity of side effects (mild, moderate or severe) for each side effects (gastrointestinal issues, headaches, fatigue or dizziness, mood changes)? Is there a structured questionnaire for participants to fill in?

Response:

Thank you for this requesting these additional details. The table (Table 5) now includes a severity level breakdown across each side effect type. At program initiation patients are clearly instructed to provide a simple description any side effects that arise and to give the event a severity rating. This detail has been added to the relevant paragraph of the program overview section (Line 189-191), as follows:

“Patients are instructed to report side effects whenever they manifest by emailing their MDT with a simple description of the event and giving a severity rating (mild, moderate, or severe).“

Comment:

  1. Discussion, please refer to my major comment.

Response:

As above.

Reviewer 2 Report

Comments and Suggestions for Authors

The authors present a novel study investigating the role of DWLS in promoting weight-based health outcomes in individuals taking GLP-1 agonists. While previous studies have been conducted in other continents the authors present a novel study investigating the impact of DWLSs in Europe.

The manuscript could be improved with the following edits:

-            The authors mention OECD countries, please write out this acronym for the audience

-            One significant limitation to the Juniper platform is the lack of a registered dietitian on the healthcare team, this should be noted in the manuscript, one would assume more pronounced results with a qualified health professional in the nutrition field

-            It would be interesting and helpful for practitioners to note if side effect rates changed throughout the course of the 6 months, ie did side effects decrease as time on the medication went on?

Author Response

Reviewer 2:

The authors present a novel study investigating the role of DWLS in promoting weight-based health outcomes in individuals taking GLP-1 agonists. While previous studies have been conducted in other continents the authors present a novel study investigating the impact of DWLSs in Europe.

Comment:

The manuscript could be improved with the following edits:

-            The authors mention OECD countries, please write out this acronym for the audience

Response:

Thank you for noticing this. We have now written out the acronym in full.

Comment:

-            One significant limitation to the Juniper platform is the lack of a registered dietitian on the healthcare team, this should be noted in the manuscript, one would assume more pronounced results with a qualified health professional in the nutrition field

Response:

Thank you for identifying the importance of this. We apologize for not being specific in the program overview section and have now changed the MDT description to the following:

“…they are allocated an MDT, consisting of a doctor, a university-qualified nutritionist (hereafter referred to as a health coach), and a nurse practitioner.”

We wanted to use the term ‘health coach’ in the rest of the article to keep the language consistent with previous publications, however, we are more than happy to substitute each subsequent reference to ‘health coaches’ with the word ‘nutritionist(s)’ if you feel this is necessary.

Comment:

-            It would be interesting and helpful for practitioners to note if side effect rates changed throughout the course of the 6 months, ie did side effects decrease as time on the medication went on?

Response:

  • Thank you for this excellent suggestion. Unfortunately we (study investigators) do not have access to exact side effect dates for this cohort, so we cannot report on the temporal pattern of side effect incidence. However, we have passed on this feedback to the Eucalyptus clinical auditing team to ensure that it becomes accessible through the central data repository for future research.

Round 2

Reviewer 2 Report

Comments and Suggestions for Authors

Thank you for your thoughtful responses and edits to the manuscript.